

# Enhanced functional connectivity properties of human brains during in-situ nature experience

Zheng Chen[1],[*], Yujia He[2],[*] and Yuguo Yu[2]

[1] Key Laboratory of Ecology and Energy-Saving Study of Dense Habitat, Department of Landscape Studies, College of Architecture and Urban Planning, Tongji University, Shanghai, China
[2] The State Key Laboratory of Medical Neurobiology and Institutes of Brain Science, School of Life Science and the Collaborative Innovation Center for Brain Science, Center for Computational Systems Biology, Fudan University, Shanghai, China
[*] These authors contributed equally to this work.

## ABSTRACT

In this study, we investigated the impacts of in-situ nature and urban exposure on human brain activities and their dynamics. We randomly assigned 32 healthy right-handed college students (mean age = 20.6 years, SD = 1.6; 16 males) to a 20 min in-situ sitting exposure in either a nature (n = 16) or urban environment (n = 16) and measured their Electroencephalography (EEG) signals. Analyses revealed that a brief in-situ restorative nature experience may induce more efficient and stronger brain connectivity with enhanced small-world properties compared with a stressful urban experience. The enhanced small-world properties were found to be correlated with "coherent" experience measured by Perceived Restorativeness Scale (PRS). Exposure to nature also induces stronger long-term correlated activity across different brain regions with a right lateralization. These findings may advance our understanding of the functional activities during in-situ environmental exposures and imply that a nature or nature-like environment may potentially benefit cognitive processes and mental well-being.

## INTRODUCTION

Evidence has suggested that, compared to urban exposure, nature exposure may have a restorative effect on human health and well-being (*Bowler et al., 2010*; *Calogiuri & Chroni, 2014*; *Groenewegen et al., 2006*; *Velarde, Fry & Tveit, 2007*), especially in regulating emotion (*Capaldi, Dopko & Zelenski, 2014*) and mitigating excessive arousal (*Jiang, Chang & Sullivan, 2014*; *Li & Sullivan, 2016*; *Ulrich, 1984*; *Ulrich, 1981*; *Ulrich et al., 1991*). More recent studies have demonstrated improvements in working memory and attention shortly after a short-term nature experience (*Berman, Jonides & Kaplan, 2008*; *Lee et al., 2015*; *Taylor & Kuo, 2009*). In one example, a brief nature walk significantly reduced anxiety-related neural activity in the subgenual prefrontal cortex (*Bratman et al., 2015b*), which enhanced cognitive performance (by increasing working memory performance)

Corresponding authors
Zheng Chen,
zhengchen@tongji.edu.cn
Yuguo Yu, yuyuguo@fudan.edu.cn

and affective regulation (by reducing negative affect and rumination while increasing positive affect). This phenomenon suggests that the natural environment may be able to influence the human cognitive state, which is less understood than emotion regulation and arousal mitigation are (*Bowler et al., 2010*).

Previous studies have suggested that the 1/f characteristics in the log-log power spectrum may be potential key factors in shaping cognitive functions during natural adaptation and evolution. The 1/f statistic describes the power composition from low to high frequencies. It is a unique statistical feature of natural signals that is widely observed in natural environments but rare in urban environments. Studies have revealed that mammalian brains can perform more efficiently in response to naturalistic signals than artificial ones (*Simoncelli & Olshausen, 2001*). A nature-like signal should contain a high-order statistic of a $1/f^\beta$-like power spectrum. In-vivo evidence has demonstrated that mammalian sensory systems (*Gal & Marom, 2013*; *Yu, Romero & Lee, 2005*) can process natural signals more efficiently than artificial signals. This characteristic of cognitive functioning may impact the efficiency of neural networks (*He, 2011*) and may eventually define global cognitive performance.

The evidence above suggests a new hypothesis: signal statistics may be one critical factor that drives the human brain to perform more efficiently in nature settings than in urban settings. Hence, in this paper, we examine the following questions: Do nature and constructed artifacts differ significantly in their statistics of visual stimuli? Do human brains respond differently to these different statistics? What are the significant differences in the brain when responding to the two types of environmental signals? Although visual signals are probably among the most important stimuli in the two types of environments, we believe that visual signals only partially capture the differences between the environments. To fully capture the holistic environmental experience, multisensory immersion is crucial. Therefore, we adopted in-situ exposure herein instead of pictorial representation.

Our new hypothesis is supported by a recent analysis of auditory perception (*Fintzi & Mahon, 2014*). A visual study of urban and nature scenes found that stress and cognitive load are more sensitive to low spatial frequencies of the scenes, whereas affective responses are more sensitive to mid-to-high spatial frequencies (*Valtchanov & Ellard, 2015*), which suggests that brain functions may be sensitive to frequencies. Another study found that, compared to random noise, music sound with a 1/f property can induce enhanced brain connectivity with efficient information flow across brain regions featuring a small-world complex network property (*Wu et al., 2013*; *Wu et al., 2012*). Small-world networks are hierarchical structures with more efficient and well-connected hubs, which are widely found in biological, ecological, social, world-wide web, molecular and neuronal networks (*Watts & Strogatz, 1998*). Because of these hubs, the small-world network usually entails a large clustering coefficient and short path length, which enables a more efficient flow of information than that found in randomly ordered nonhierarchical networks.

To answer the questions above, we collected data both from brain neural activities and from subjective perceived experience using a portable electroencephalogram (EEG) device and psychological scales, respectively. We then examined the EEG functional connectivity during in-situ nature and urban experiences, including functional

correlation, small-world network statistics, 1/f statistics and their lateralization, and compared EEG statistics with a subjective experience measured by psychological scales.

## METHODS

### Subjects

In this study, we recruited 32 healthy, right-handed participants (mean age = 20.6, SD = 1.6, 16 males) from among Chinese college students. A between-subjects design was used in this study, in which participants were randomly assigned to a nature environment (n = 16, 6 males) or an urban environment (n = 16, 10 males). This study was approved and supervised by the Ethics Committee of Tongji University (no. 2015yxy103).

No significant levels of neuroticism were detected among the participants as indicated by the scores (Table 1) on the neuroticism subscale of the NEO Personality Inventory (*Costa & McCrae, 1992*). All participants reported a small amount of daily stress and had relatively stress-reducing living environments. The subjects were generally well-rested with a broad range of work and entertainment schedules; however, all subjects participated in moderate exercise and were exposed to nature on a daily basis.

### Environments

Because human cognition of nature and the urban environment may be sensitive to the frequencies of visual stimuli (*Valtchanov & Ellard, 2015*), we intentionally controlled the proportion of frequencies and used the $\beta$ values in the 1/f statistics of visual stimuli as the criteria for site selection. Research has shown that the $\beta$ values of nature images are typically close to 2 (*Szendro, Vincze & Szasz, 2001*; *West & Shlesinger, 1990*) due to an appropriate distribution between low-frequency contours (e.g., shapes of trees or mountains) and high-frequency details (e.g., fractal edges, random texture and lines). This finding implies a moderate long-term correlation level across image components of all frequency ranges. With increasingly uniform materials (e.g., concrete) and limited high-frequency detail, urban scenes may be more likely to reveal a larger $\beta$, but we were less confident about this hypothesis before verification.

To find a typical nature and urban environment, we first examined a large number of photographs. Because of limited confidence in the $\beta$ estimation for urban scenes, we intentionally investigated a larger sample of photographs of urban scenes (n = 135) than nature scenes (n = 80). All of the photos had been captured and reviewed by professional landscape architects and architects. Both nature and urban photos revealed a normal distribution of $\beta$ (Figs. 1C and 1D), with means of 2.30 (SD = 0.22) and 2.61 (SD = 0.18), respectively (Fig. 1). These results were consistent with those from a previous study (*Braun et al., 2013*) that documented an average $\beta$ of 2.24 (SD = 0.19) for nature scenes and 2.53 (SD = 0.24) for buildings, which were obtained from sample sizes of 200 photos each.

Therefore, we intentionally selected two typical sites (Fig. 2) with representative $\beta$ slopes: a wooded campus garden ($\beta$ = 2.24) for the nature scene and a traffic island under an elevated highway ($\beta$ = 2.62) for the urban scene. From where the participants were seated, the nature scene consisted of 89% visible greenery and water and only 4% visible buildings and/or paved areas; the urban scene consisted of only 8% visible greenery and 56% visible

**Table 1 Personality and everyday life reported by participants before experimentation.**

| Personality and everyday-life factors | Nature (n = 16) | | Urban (n = 16) | |
| --- | --- | --- | --- | --- |
| | Mean | SD | Mean | SD |
| Age | 20.56 | 1.50 | 20.63 | 1.78 |
| NEO-PI neuroticism score* | −0.44 | 0.72 | −0.26 | 0.74 |
| Perceived level of everyday stress* | 0.13 | 1.89 | 0.56 | 1.93 |
| Perceived level of stress in living environments* | 0.56 | 1.26 | 0.06 | 1.48 |
| Total hours of sleeping per day | 6.94 | 0.93 | 6.88 | 1.64 |
| Total hours of exposure to nature per day | 0.79 | 0.56 | 0.84 | 0.79 |
| Total hours of study per day | 8.69 | 2.85 | 7.69 | 3.69 |
| Total hours of physical exercises per day | 0.64 | 0.49 | 0.81 | 0.48 |
| Total hours of entertainment per day | 3.09 | 2.44 | 3.47 | 2.28 |

Note:
* Factors were measured on a 7-point Likert scale.

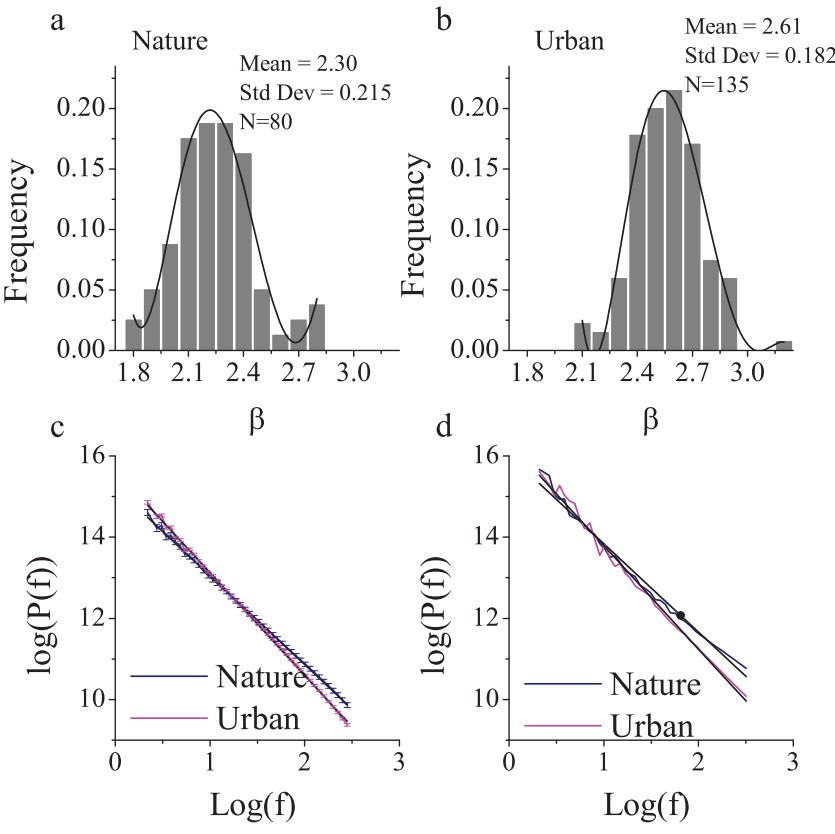

**Figure 1 The 1/f statistics of visual stimuli in nature and urban scenes.** Eighty nature images and 135 urban environment images were analyzed and yielded distinct $\beta$ slopes (A) in the nature (mean = 2.30, SD = 0.22) and (B) in the urban environment (mean = 2.61, SD = 0.18, p < 0.001). The log-log plot of frequency and power (C) revealed that nature images contain significantly more high frequencies than urban images. The two sites that we chose were of a visual 1/f statistics representative of a nature ($\beta$ = 2.24) and urban-built environment ($\beta$ = 2.62 (D)). Error bars in (C) denote SEM.
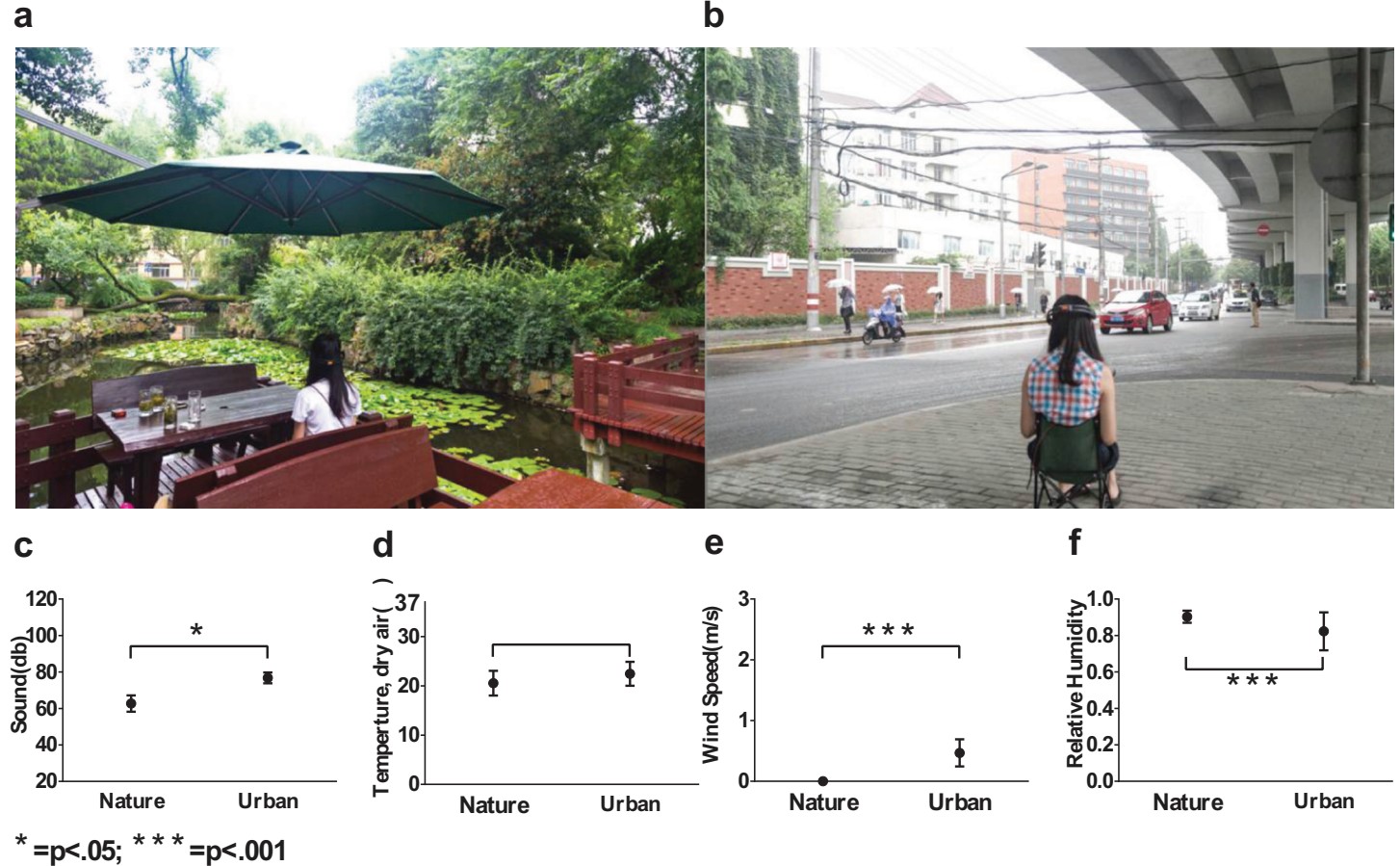

**Figure 2 Environmental conditions used in the experiment.** A shaded traffic island was used as an urban site (A) and a campus garden was used as a nature site (B), acoustic conditions (C), temperatures (D), wind speed (E) and relative humidity (F) were intentionally documented and controlled. Error bars denote SD.

buildings and/or paved areas. Based on a focus group interview after a site visit, the nature environment is highly restorative whereas the urban environment is clearly not.

In-situ environmental stimuli were adopted instead of pictorial representations to capture the multisensory experiences of the nature and urban environments. We reported possible factors that may have influenced the viewers' on-site experiences, and some of the factors were controlled (Table 2; Figs. 2C–2F). The uncontrolled factors constituted important features of the nature and urban experience.

## Measures

### EEG data collection, pre-processing and power analysis

An Emotiv wireless headset (*Dekihara & Iwaki, 2014*) and its accompanying software were used to measure EEGs. The headset consists of 14 sensors positioned on the scalp of the study subject according to the international 10–20 system: antero-frontal (AF3, AF4, F3, F4, F7, F8), frontocentral (FC5, FC6), occipital (O1, O2), parietal (P7, P8) and temporal sites (T7, T8). Brain waves were measured in terms of amplitude (10–100 microvolts) and frequency (1–70 Hz). EEG data were analyzed via Matlab and EEGlab. Data were first
| Table 2 Perceptive factors that may have impacted the experience and/or neural activities. | |
|---|---|
| **Types of perception** | **Factors that may have impacted the experience and/or the neural activities** |
| Visual | *Controlled* |
| | 1. Daylight |
| | 2. Shaded locations and illumination |
| | 3. 1/f Statistics: $\beta_{nature} = 2.30 \pm 0.22$ (SD), $\beta_{urban} = 2.61 \pm 0.18$ |
| | *Uncontrolled* |
| | 1. Motions |
| | 2. Colors |
| | 3. Types of stimuli |
| Acoustic | *Controlled* |
| | 1. Acoustic intensity: $I_{nature} = 62.70 \pm 4.50$ db, $I_{urban} = 76.80 \pm 2.97$ db |
| | *Uncontrolled* |
| | 1. Sources (The nature environment sounds consisted mostly of birds singing, humans speaking and leaves trembling; the urban environment sounds were mostly from vehicles.) |
| Haptic | *Controlled* |
| | 1. Weather: both are of a comparable mix of sunny, cloudy and rainy days. |
| | 2. Temperature: $T_{nature} = 20.53 \pm 2.50$ °C, $T_{urban} = 22.44 \pm 2.42$ °C |
| | 3. Humidity: $T_{nature} = 62.70 \pm 4.50$%, $T_{urban} = 76.80 \pm 2.97$% |
| | 4. Wind speed: $W_{nature} = 0.00 \pm 0.00$ m/s, $W_{urban} = 0.46 \pm 0.22$ m/s |
| Olfactory | *Controlled* |
| | 1. Absence of strong smell(s) |
| | *Uncontrolled* |
| | 1. Sources (e.g., gas smell in the urban environment and grass smell in nature) |
| Body movements | *Controlled* |
| | 1. Still sitting: instructions were given before the exposures that participants should sit as still as possible. |
| | *Uncontrolled* |
| | 1. Eye movements |

filtered (< 0.5 Hz or > 50 Hz) using EEGlab and artifacts were removed using Adjust 1.1 (*Mognon et al., 2011*).

The signal was first analyzed via fast Fourier transformation and then categorized into five frequency bands: delta (1–4 Hz), theta (5–8 Hz), alpha (9–12 Hz), beta (13–25 Hz), and gamma (26–45 Hz). The total mean power was calculated by the frequency bands and brain connectivity.

### EEG functional connectivity

To evaluate EEG functional connectivity, we first calculated the EEG correlations and then calculated the small-world network statistics.

*Correlations*

Correlations between signals or signal frequency components at different electrodes were calculated using the following equation:

$$r(x) = C_{AB}(x)/(C_{AA}C_{BB}) \tag{1}$$

where the cross-covariance between signals A and B was noted as $C_{AB}$ and the auto-covariances of signals $A$ and $B$ were noted as $C_{AA}$ and $C_{BB}$, respectively (*Guevara & Corsi-Cabrera, 1996*).

*Small-world network statistics*

Small-world networks are hierarchical structures with more efficient and well-connected hubs, which are widely found in biological, ecological, social, world-wide web, molecular and neuronal networks (*Watts & Strogatz, 1998*; *Wu et al., 2013*). A small-world network is defined by two measures: the clustering coefficient ($C_{mean}$) to describe functional segregation and the average shortest path length ($L_{mean}$) to describe functional integration (*Rubinov & Sporns, 2010*). A small-world network is usually more efficient (*Honey et al., 2009*) because it is highly segregated (larger $C_{mean}$) and highly integrated (shorter $L_{mean}$). The former allows divisions for tasks to be specialized while the later allows coordination to support collaboration between divisions.

Based on correlations, we measured small-world network propriety using $L_{mean}$, which is the average shortest path length, and $C_{mean}$, which is the clustering coefficient. The clustering coefficient of an electrode $v_i$ was calculated as follows:

$$C_{v_i} = \begin{cases} 0 & k \leq 2 \\ \frac{2n}{k(k-1)} & k > 2 \end{cases} \tag{2}$$

where $k$ denotes the number of electrodes of an EEG correlation with the electrode $v_i$ higher than a given threshold, and n denotes the number of paths with a correlation higher than the threshold between the connected $k$ electrodes (where $n$ equals $C_k^2$ in a saturated scenario). The average clustering coefficient for a participant was then calculated as follows:

$$C_{mean} = \frac{1}{14}\sum_{i=1}^{14} C_{v_i} \tag{3}$$

The shortest path length was calculated from the shortest of all total paths from $v_i$ to $v_j$ for a pair of electrodes $v_i$ and $v_j$ using the Floyd algorithm (*Floyd, 1962*). The Floyd algorithm defines a $n \times n$ weight matrix as

$$D = \left(d_{ij}\right)_{n\times n} := \begin{bmatrix} d_{11} & d_{12} & \ldots & d_{1n} \\ d_{21} & d_{22} & \ldots & d_{2n} \\ \ldots & \ldots & \ldots & \ldots \\ d_{n1} & d_{2n} & \ldots & d_{nn} \end{bmatrix}, \quad d_{ij} \in R[0, +\infty) \tag{4}$$

in which

$$d_{ij} = \begin{cases} 0, & i = j \\ 1, & r_{ij} \geq Threshold \\ 13, & r_{ij} \leq Threshold \end{cases} \tag{5}$$

where $r_{ij}$ denotes the maximum from the correlation function between electrodes $v_i$ and $v_j$ with phase difference considered in the calculation, while $d_{ij}$ denotes the connectivity distance instead of the physical Euclidean distance. The weight matrix $D$ is calculated as follows:

1. First an initial matrix was defined as $D^{(0)} = D$;
2. Then, let $D^{(k)} = \left( d_{ij}^{(k)} \right)_{n \times n}$, $k = 1, 2, \ldots, n$, where

$$d_{ij}^{(k)} = \min\{ d_{ij}^{(k-1)}, d_{ik}^{(k-1)} + d_{kj}^{(k-1)} \} \tag{6}$$

3. Then $D^{(n)} = \left( d_{ij}^{(n)} \right)_{n \times n}$, where $d_{ij}^{(n)}$ is the shortest path from electrode between electrodes $v_i$ and $v_j$.

The path length between two immediate neighbors was coded as 0 (to itself), 1 (beyond threshold) or 13 (below threshold). By definition, the weight $D$ is usually calculated using the positive infinity of a comparable disconnected network. However, this calculation may be flawed because the average of the shortest path length is oversensitive to the isolated points. Only one or two isolated points will drive the average to positive infinity when the whole network is far from being truly disconnected. To correct this oversensitivity, we used a large number instead of positive infinity. We interpreted the disconnected electrodes $v_i$ and $v_j$ as a situation when $v_i$ must travel through all other electrodes to reach $v_j$, which is mathematically 13 (or the number of electrodes (14) minus one). For each participant,

$$D_{mean} = \frac{1}{C_{14}^2} \sum_{\substack{i=1, j=1 \\ i<j}}^{14} d_{v_i, v_j} \tag{7}$$

where $L_{vi, vj}$ denotes the shortest path length between a pair of electrodes $v_i$ and $v_j$. For comparison purpose with networks with different number of electrodes, we normalized $D_{mean}$ as $L_{mean}$, where $L_{mean} = \frac{D_{mean}}{Electrodes\_Number - 1} = \frac{D_{mean}}{13}$.

*Random network statistics*

It is a standard procedure in small-world network analysis to report random referencing networks for control purposes. We used the following methods to construct such random networks and to calculate corresponding $L_{mean}$ and $C_{mean}$:

1. For a participant $m$, we first calculated the EEG correlation matrix $R_m := \left( r_{ij} \right)_{14 \times 14}$.
2. We transformed $R_m$ to a binary (0–1) adjacent matrix $A_{m,th} := \left( a_{ij} \right)_{14 \times 14}$, where for a threshold of $th$, $a_{ij} = \begin{cases} 1, & r_{ij} \geq th \\ 0, & r_{ij} < th \end{cases}$.

3. We counted the number of cells with a value of 1 in the upper non-diagonal half of $A_{m,th}$, that is, when $a_{ij} = 1 (i < j)$ as $n_a$. The number of such non-diagonal cells in $A_{m,th}$ should be $2 \times n_a$ because of symmetry. Because $R_m$, and hence $A_{m,th}$, was diagonally symmetric, only the cell value of the upper half matrix needed to be assigned.

4. We counted the total number of isolated points in $A_{m,th}$ as $n_{iso}$. If all cells in the $i$th row and $i$th column except the diagonal cell $a_{ii}$ are 0, the $i$th electrode is defined as an isolated point ($i = 1, 2, \ldots, 14$).

5. We constructed a comparable random adjacent matrix $A_{m,th}^{Rand} = \left(a_{ij}^r\right)_{14 \times 14}$ for $A_{m,th}$. We first defined diagonal cells as $a_{ii}^r = 2$ and constructed $n_{iso}$ isolated points by randomly assigning 0 to all the cells in the same row or column of $n_{iso}$ number of diagonal cells. Then we randomly assigned $n_a$ number of 1 to the upper non-diagonal half matrix of $A_{m,th}$, except those that were already defined as 0. All undefined cells in the upper non-diagonal half matrix were assigned as 0.

6. We then calculated the random adjacent matrix of clustering coefficients $CC_{m,th}^{rand} = \left(cc_{ij}^r\right)_{14 \times 14}$, letting $cc_{ij}^r = a_{ij}^r$. $cc_{ij}^r$ was denoted as 1 when electrodes $v_i$ and $v_j$ were connected or 0 if electrodes were not counted.

7. We then calculated the random adjacent matrix of the shortest path length

$$D_{m,th}^{rand} = \left(d_{ij}^r\right)_{14 \times 14} \text{ where } d_{ij}^r = \begin{cases} 0, & a_{ij}=2 \\ 1, & a_{ij}=1 \\ 13, & a_{ij}=0 \end{cases}.$$

*Power spectrum density 1/f statistics*

The power spectral density $P(f)$ describes how the power of a signal $x(t)$ or a time series is distributed with frequency. It is usually calculated by the following Fourier transform equations. First, the Fourier transform of a time domain signal $x(t)$ in frequency domain $x(f)$ is calculated as

$$x(f) = \int_{-\infty}^{\infty} x(t) e^{-2\pi i f t} dt \tag{8}$$

where $f$ is the frequency in Hz, i.e., cycles per second. The integrand $|x(f)|^2$ can be interpreted as an energy density function describing the energy per frequency unit contained in the signal at the frequency $f$. The power spectrum density of a signal $x(t)$ is thus defined as

$$P(f) = \lim_{T->\infty} \frac{|x(f)|^2}{T} \tag{9}$$

where T is the infinite time.

The 1/f statistic or $1/f^\beta$, which is a log-log linear correlation between power spectrum and frequency, is commonly found in natural signals. 1/f statistics of EEG networks describe the nested frequency characteristic of brain networks. Previous evidence has indicated that a larger $\beta$ is found in human brain networks during task-free rest, while $\beta$
decreases during tasks (*He et al., 2010*), and more difficult tasks have been found to trigger larger decreases in $\beta$ (*Ward, 2002*).

The 1/f statistics were calculated as the regression slope $\beta$ (absolute value) of the log power over the log frequency in a $1/f^{\beta}$ power spectrum. The $\beta$ values were first calculated at each electrode for each individual participant and then linearly combined as electrode and individual global means. Lateralization was calculated as log(R)-log(L) where R and L represent the right and left symmetrical pairs (i.e., AF4/AF3, F8/F7, F4/F3, FC6/FC5, T8/T7, O2/O1), respectively.

### Subjective affect, experience and attention

We used a 40-item Profile of Mood States (POMS) scale (*Grove & Prapavessis, 1992*) to measure the affective states before and after environmental exposure. Participants completed a POMS scale upon arrival at either experimental site and completed another scale after a 20-min exposure at the respective site. By linearly combining the 40 items, the scale reveals eight affective measures (tension, anger, fatigue, depression, vigor, esteem, confusion, and total mood disturbance (TMD)). We calculated the changes of the eight measures, which revealed the impact of environmental exposure on subjective affect, and compared the differences in changes between nature and urban groups.

We used a revised Perceived Restorativeness Scale (PRS) to measure participants' perception about their environmental experience. *Kaplan (1995)* summarized the four key qualities of environmental visual stimuli that induced a restorative experience: (1) the environment should offer a "being away" feeling, i.e., a mental refuge from everyday chores and routines; (2) the environment should induce mild and positive emotional arousal that he termed "(soft) fascination"; (3) the environment should offer "coherent" information for "extent," which should be comprehensible and should allow fruitful explorations, independent of complexity; (4) the environment should promote "compatibility," i.e., a potential to afford certain desirable activities. Based on Kaplan's theory, *Hartig et al. (1997)* developed the PRS to measure the restorative qualities of nature visual exposure. We collected feedback from a focus group of participants with comparable backgrounds and revised the scale to better capture the experience reported by the focus group (Table 2).

We used the Necker Cube Pattern Control Test to measure participants' attention change (*Orbach, Ehrlich & Heath, 1963*) before and after environmental exposures. A higher attention level is indicated by a smaller number of cube pattern changes. We used the change before and after exposure $N_{post} - N_{pre}$ to measure the impact of environmental exposure on one's attention.

### Statistical test

We conducted both a participant-wise comparison using independent t-tests and an electrode-wise comparison using t-tests that were paired by electrodes. If the differences were significant in both cases, only the participant-wise results were reported. Two-way ANOVA was used when two-factor comparisons were made.
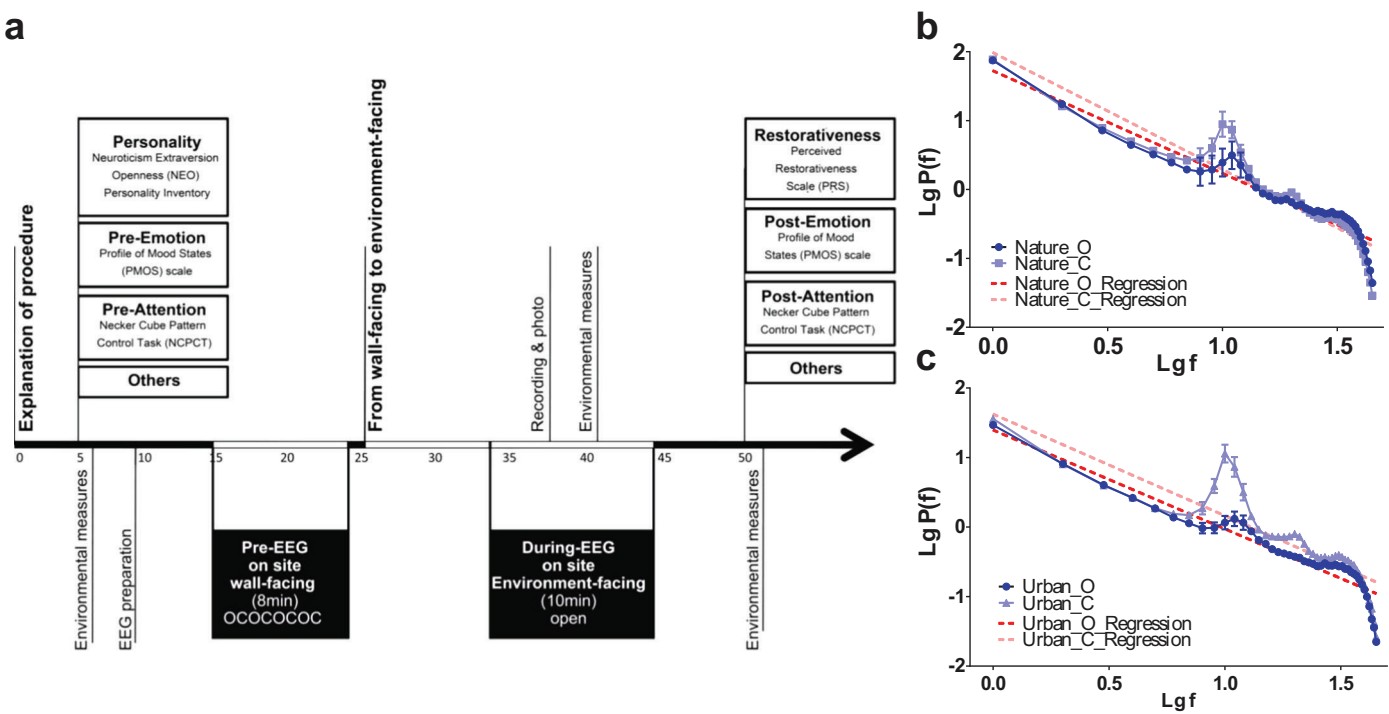

**Figure 3 Experimental procedures and data validity.** Experimental procedures (A) was illustrated. The EEG log-transformed power distribution, as observed at the occipital lobe (O1 and O2) during the pre-test eye-open sessions and compared to the eye-closed session in both the nature (B) and built (C) environment sites, revealed a clear 1/f statistic and alpha-blocking effect. Error bars denote SEM.

## Procedure and data validity control

The experiment was performed via a structured procedure (Fig. 3). The 32 participants were first randomly assigned to either a nature exposure group (n = 16) or an urban exposure group (n = 16). After first signing a written consent form containing an explanation of the experimental procedure and risks, participants completed a two-page pre-test questionnaire followed by the Necker Cube Pattern Control Test. Portable EEG electrodes were positioned onto the scalps of participants, who were then asked to sit facing a wall to temporarily exclude visual information. Participants were instructed to alternate between either opening or closing their eyes in four one-minute cycles. After this pre-experimental phase, participants were told to turn and face the environment for 20 min.[1] Participants were asked to sit facing either an urban environment (i.e., a traffic island under an elevated highway) or a natural environment (i.e., a heavily wooded campus garden). After viewing the environment, participants completed a posttest questionnaire, a short interview and Necker cube test again. To prevent their minds from wandering or becoming drowsy, participants were told to count from 1 to 1,000 soundlessly at a speed that was slow enough not to interrupt the environmental experience.

We specifically adopted eye-opened and eye-closed pretests in the beginning of the experiment to verify the reliability/stability of electrode recording across the 20 min of the environmental exposure. We examined the power distribution and 1/f statistics of two

[1] EEGs of the entire 20-min exposure were recorded for the latter 16 participants (8 nature) while only the latter half (10 min) of the exposure was measured for the first 16 participants (8 nature). Paired t-tests revealed no significant differences between the first half (10 min) and the latter half (10 min) of exposure for the 16 participants whose EEGs were recorded for the entire 20 min in terms of EEG correlation and $\beta$ exponent, except for two participants in the urban environment groups. Therefore, the full recording length from both groups (10 min for the first half and 20 min for the latter half) was used for analyses.

electrodes at the primary visual cortex (O1 and O2) during the two pretests. The results revealed a clear 1/f statistic in both the nature and urban environments. At both sites, there was also a clear bump at the top of the 1/f slope near logf = 1 (the location of the alpha range) during the eye-closed session, while the bump dropped towards the slope during the eye-open session (Figs. 1B and 1C). This bump-dropping effect during the eye-open session is called alpha-blocking (*Könönen & Partanen, 1993*). The successful capture of 1/f statistics (arrhythmic firing) and alpha-blocking (rhythmic oscillation) supports the validity of the data.

# RESULTS

## EEG functional connectivity

### *Signal correlation*

We computed the functional connectivity by calculating both the time domain EEG correlation (Figs. 4A and 4B) across electrodes and the frequency domain correlation (Figs. 4C and 4D). The EEG component correlation across recording sites for the theta frequency band is shown (the other delta, alpha and beta bands have similar results and were not shown here). The direct EEG amplitude correlation across recording sites demonstrated a higher correlation on the right side of the brain than on the left side, suggesting more synchronized EEG signals on the right side. Additionally, the correlation on the right side of the brain is higher for natural exposure than in the urban environment. The EEG measure of frequency-dependent functional connectivity reveals the coherent property of information transfer at different frequency ranges across recording regions (*Fries, 2005*; *Varela et al., 2001*). A participant-wise two-way ANOVA was performed on the mean EEG frequency correlation of 14 electrodes and at two conditions (nature and urban environment exposures). We observed a significantly higher correlation (in frequency domain, see Figs. 4C and 4D) of overall electrodes during nature exposure compared to that during urban environment exposure (the ratio of between-group difference over within-group difference was $F(1,420) = 14.68$, $p < 0.001$), while no significant differences were found in the wall-facing eye-open baselines ($p = 0.314$). Stronger functional connectivity networks were found in the right hemisphere during the nature exposure than the urban exposure. An electrode-wise two-way ANOVA was then performed on the EEG power correlations in the four conventional frequency bands (delta, theta, alpha and beta) and under two conditions (nature and urban environment exposures), which was followed by a t-test paired by electrodes. We observed a significantly higher overall power correlation during nature exposure compared to that during urban environment exposure ($F(1,134) = 120.14$, $p < 0.001$) with a higher power correlation observed during nature exposure observed in delta ($T(13) = 10.76$, paired by electrodes, $p < 0.001$), theta ($T(13) = 6.73$, paired by electrodes, $p < 0.001$), alpha ($T(13) = 6.25$, paired by electrodes, $p < 0.001$) and beta ($T(13) = 5.06$, paired by electrodes, $p < 0.001$) frequencies.

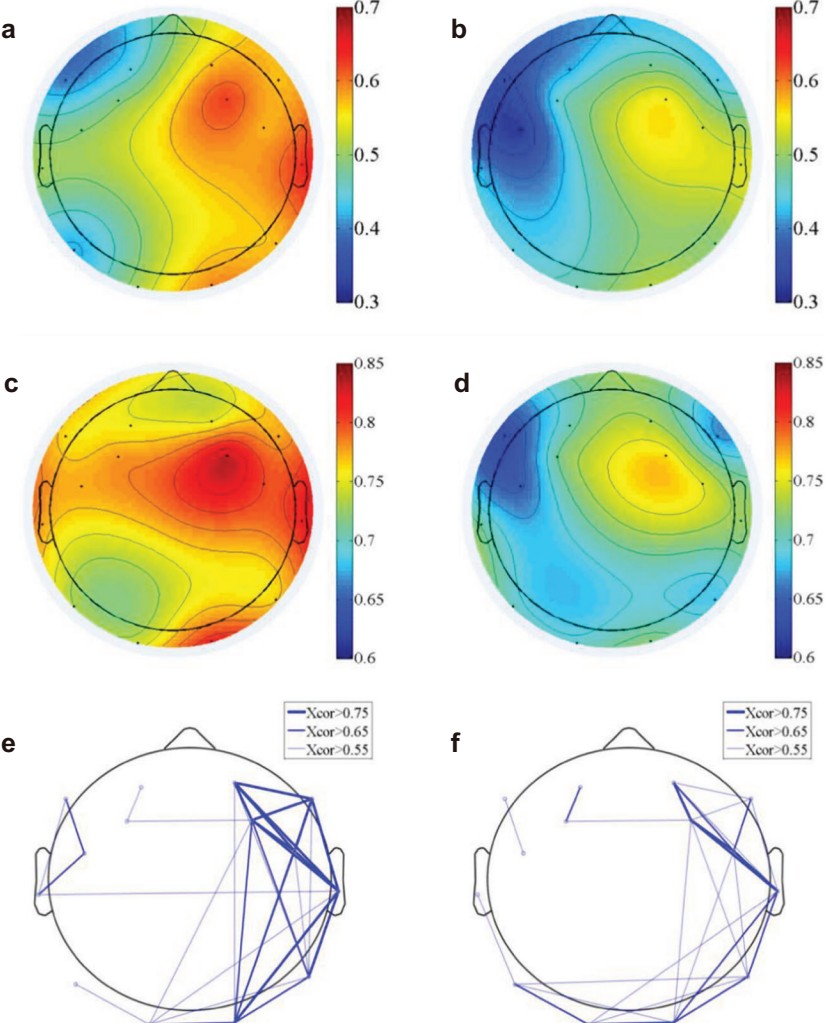

**Figure 4 EEG functional connectivity.** The time-domain functional topography during the nature (A) and urban exposures (B), which revealed a higher global EEG correlation during nature than during urban exposure (colorbar, correlation in unitless scale). The frequency-domain functional topography in the theta band revealed similar results during nature (C) and urban exposure (D, colorbar, correlation in unitless scale). The functional connectivity density were presented in the supplementary figures (Fig. S1). The functional connectivity networks revealed stronger networks in the right hemisphere during the nature exposure (E) than during the urban exposure (F).

### Small-world network

We analyzed the small-world network statistics of the EEG activity by calculating the average shortest path length ($L_{mean}$) and network clustering coefficient ($C_{mean}$). We reported $L_{mean}$ and $C_{mean}$ at different thresholds ranging from 0.05–0.95 (Fig. 5).

The average shortest path lengths ($L_{mean}$) are smaller during the nature experience than during the urban experience, but no significant difference was found except at both ends of the threshold spectrum. Both EEG networks had approximately the same value of $L_{mean}$ at their random references. The clustering coefficients ($C_{mean}$) were significantly

 

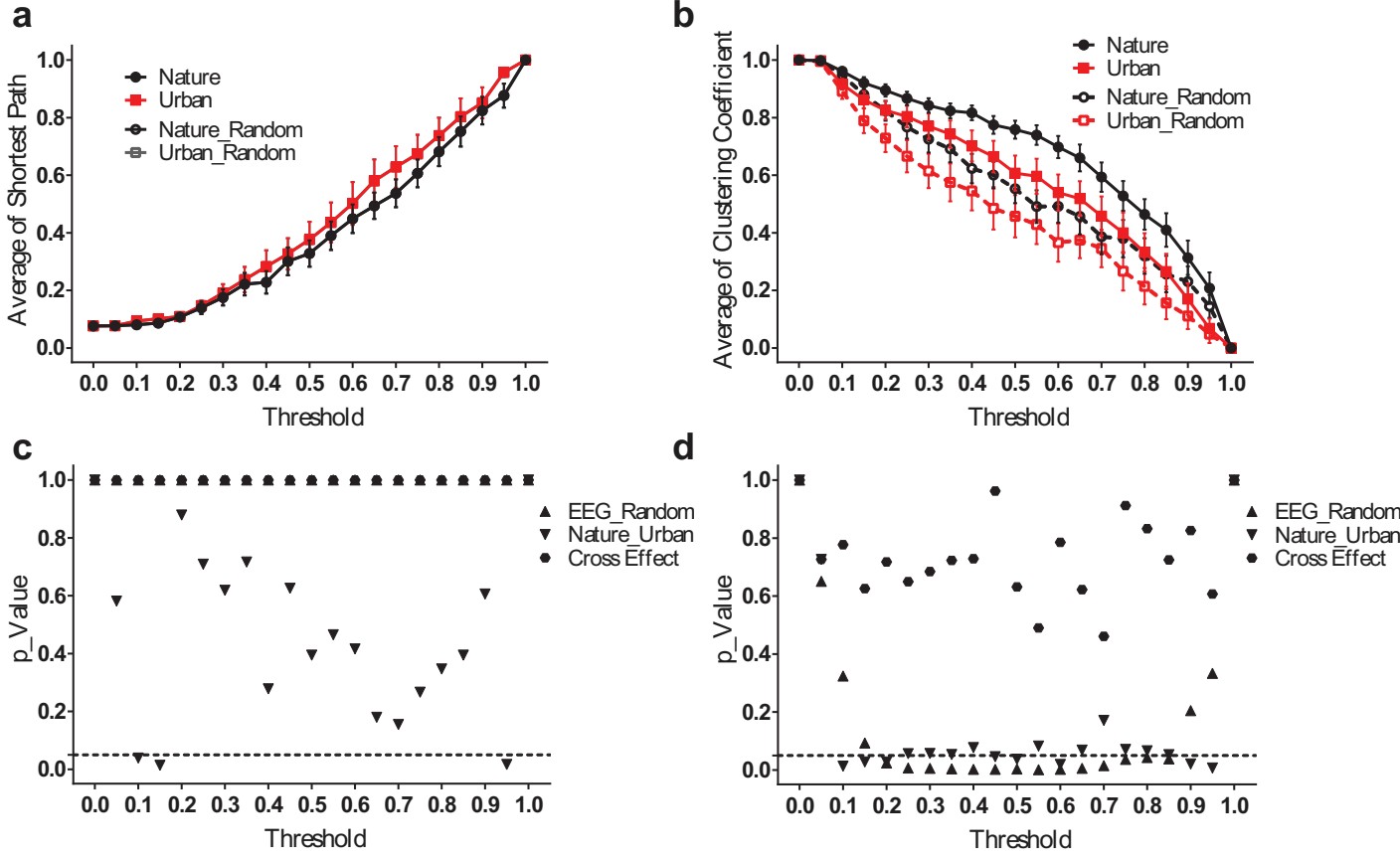

**Figure 5 EEG small-world network statistics.** The average shortest path lengths ($L_{mean}$) of EEG networks during nature and urban exposure at different thresholds were illustrated in (A), where EEG network and its random referencing network had identical $L_{mean}$. The clustering coefficients ($C_{mean}$) of EEG networks during nature and urban exposure at different thresholds were illustrated in (B), with random references. Error bars in (A) and (B) denote SEM. A two-way ANOVA was conducted to compare the difference between nature and urban sites, to compare the difference between EEGs and random references and to compare the interactions for $L_{mean}$ (C) and $C_{mean}$ (D).

higher during the nature experience than the urban experience ($p < 0.05$ for most threshold values, except 0.05, 0.40, 0.55, 0.70–0.80). At both environmental conditions, the EEG networks had a higher $C_{mean}$ than did the random references ($p < 0.05$ for most threshold values from 0.20–0.85).

However, the difference between the nature and urban environments was that the average shortest path length ($L_{mean}$) was more significant when analyzed in the frequency domain. The results for the theta band, for example (Fig. 6), revealed that the EEG functional connection during the nature exposure had a smaller average shortest path length ($p < 0.05$ from a threshold of 0.45–0.80) and a larger clustering coefficient ($p < 0.05$, from a threshold of 0.40–0.70). The results at the other frequency bands were reported in the supplementary figures (Figs. S2–S4).

### EEG 1/f characteristic

To investigate the environmental impact on the temporal statistics of the EEG recordings, we performed a Fourier transform for the EEG signals and calculated the power spectrum

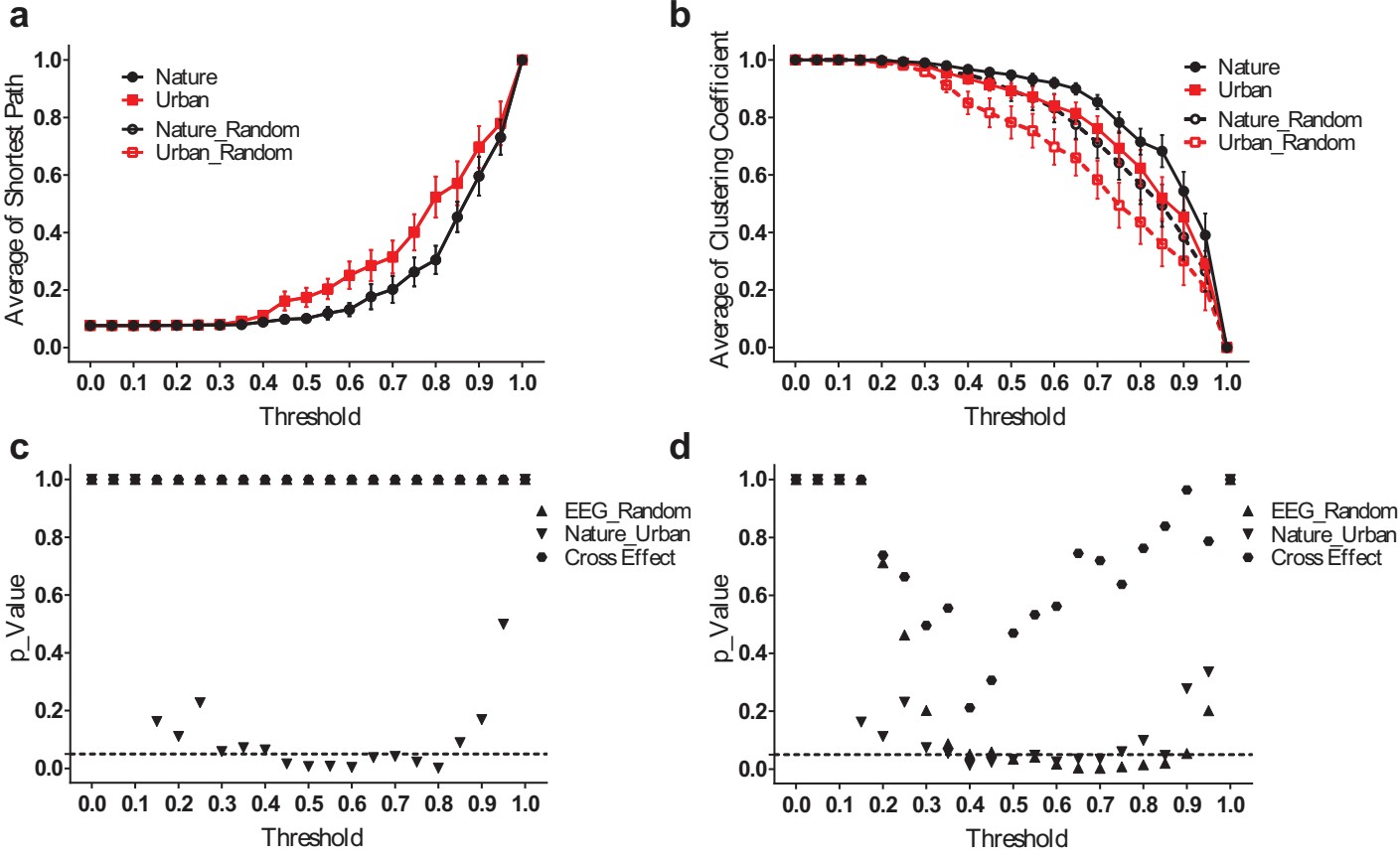

**Figure 6 EEG small-world network statistics in the theta band.** The average shortest path lengths ($L_{mean}$) of EEG networks during nature and urban exposure at different thresholds were illustrated in (A), where EEG network and its random referencing network had identical $L_{mean}$. The clustering coefficients ($C_{mean}$) of EEG networks during nature and urban exposure at different thresholds were illustrated in (B), with random references. Error bars in (A) and (B) denote SEM. A two-way ANOVA was conducted to compare the difference between nature and urban sites, to compare the difference between EEGs and random references and to compare the interactions for $L_{mean}$ (C) and $C_{mean}$ (D).

versus frequency in a log-log space. Interestingly, the EEG power spectrum also displays 1/f properties similar to those observed for the natural signals. The fitting slope, i.e., the $\beta$ value, for each recording site is approximately 1.5, and the average value for all recording sites in the natural scene is $1.62 \pm 0.027$, which is significantly higher than that in the urban environment ($1.54 \pm 0.033$, $T(13) = 2.68$, $p = 0.019$, t-test paired by electrode, Fig. 7A).

To further investigate the spatial variation of 1/f statistics, six brain regions were studied: the left (AF3, F7 and F3) and right antero-frontal (AF4, F8 and F4); left (FC5 and T7) and right temporal/frontocentral (FC6 and T8); and left (P7 and O1) and right parietal/occipital (P8 and O2). A trend towards larger $\beta$ slopes in these brain regions was observed during the nature experience (Fig. 7B). Additionally, for both conditions, $\beta$ values were higher on the right brain side than the left side, indicating a clear lateralization effect. The lateralization was quantified as $\Delta\beta = \beta_{right} - \beta_{left}$. Fig. 7C shows $\Delta\beta$ of six brain regions both for the nature and urban environments. Lateralization is

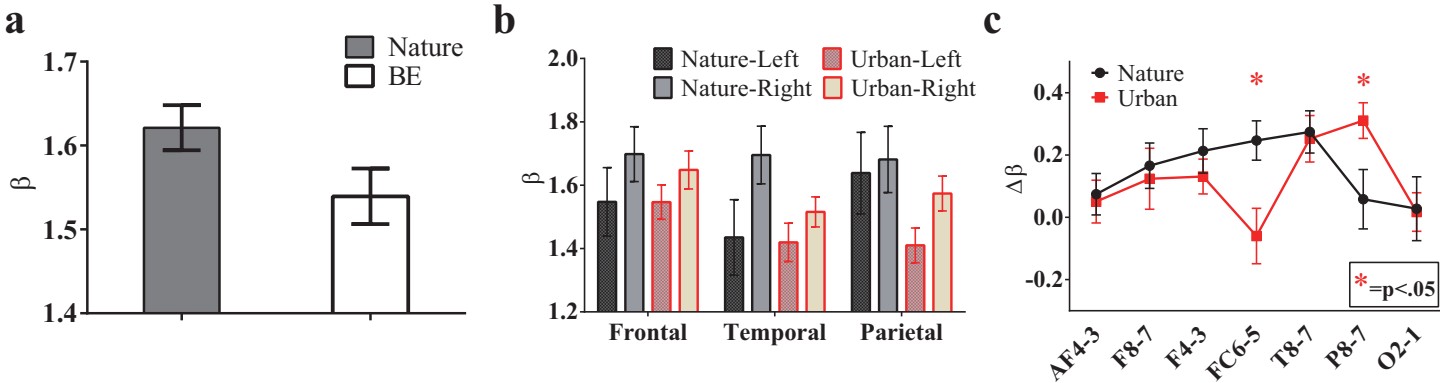

**Figure 7 The 1/f statistics of EEGs.** EEG signals during the nature experience revealed a larger global $\beta$ exponent than the EEG signals of the urban experience (A, p = 0.019, t-test paired by electrode). The $\beta$ values in the sub-brain regions are illustrated (B) The lateralization of each electrode pair (calculated by $\beta$ lateralization, i.e., $\beta$right-$\beta$left) was reported (C) with a larger right lateralization at FC6/FC5 (p = 0.009) during the nature experience and a larger right lateralization at P8/P7 (p = 0.031) during the urban experience. Error bars denote SEM.

stronger for the nature condition than for the urban condition for most brain regions except P8/P7 and O2/O1 regions.

## Subjective measures and their correlation with functional connectivity

### Perceived restorative experience and small-world network property

Participants reported a more restorative experience after a nature exposure than that after an urban exposure. After a nature exposure, participants reported higher fascination (T(28.93) = 3.23, p = 0.003), higher coherence (T(30) = 3.00, p = 0.005), increased perception of being away (T(30) = 4.31, p < 0.001) and higher compatibility (T(30) = 6.68, p < 0.001). Our observations herein are consistent with previous experimental reports (*Berto, 2007*; *Hartig et al., 1997*; *Herzog, Maguire & Nebel, 2003*).

To investigate the impact of EEGs on perceived experience, we then compared the EEG correlations during exposure with the four PRS factors and the changes of the eight POMS scores using Pearson's correlation. Moderate correlations were observed between the coherence score from the self-reported PRS factors and the average EEG correlations in the theta (correlation coefficient, i.e., cc = 0.46, p = 0.008), alpha (cc = 0.37, p = 0.040), beta (cc = 0.35, p = 0.049) and gamma (cc = 0.36, p = 0.045) bands.

We then regressed the PRS coherence scores on the small-world network statistics in the lower bands (delta, theta, and alpha, Fig. 8). The participant-wise small-world network statistics were calculated by averaging the values at mid-range thresholds from 0.65 to 0.85. Regression revealed a significant correlation between the PRS coherence scores and the individual shortest path length mean ($L_{mean}$) in theta ($R^2$ = 0.291, p = 0.001) and alpha bands ($R^2$ = 0.292, p = 0.001). There was also a significant correlation between the PRS coherence scores and the individual clustering coefficient mean ($C_{mean}$) in the theta ($R^2$ = 0.201, p = 0.010) and delta bands ($R^2$ = 0.148, p = 0.030).

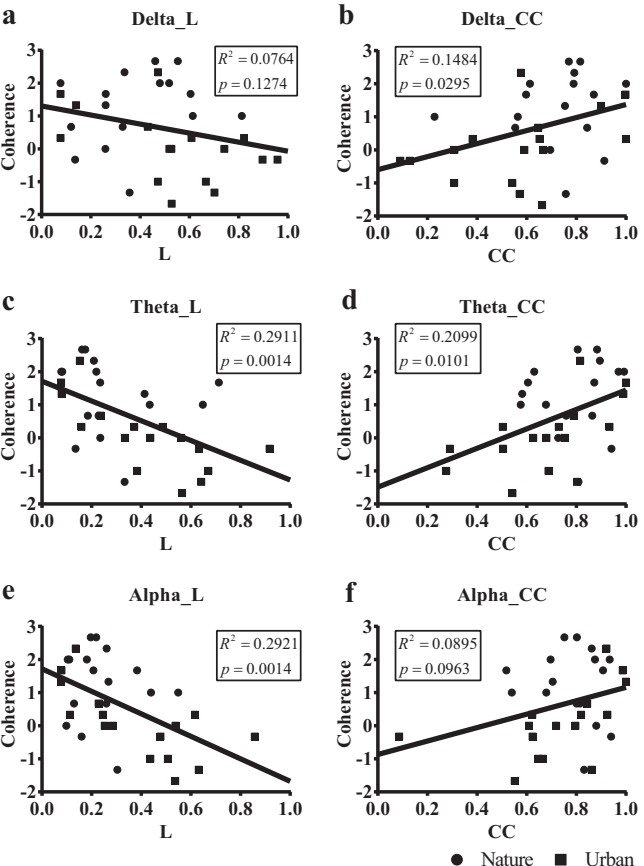

**Figure 8 Scatterplots of the PRS coherence scores of 32 participants on their small-world network statistics.** We calculated the individual shortest path length mean ($L_{mean}$, L) and the individual clustering coefficient mean ($C_{mean}$, CC) of the 32 participants by averaging the values at mid-range threshold (from 0.65 to 0.85) at different frequency bands and used the frequency bands as a predictor of PRS coherence in a simple linear regression. The x axis in (A), (C) and (E) denotes the participant's shortest path length mean (L) averaged by the 14 electrodes, while the x axis in (B), (D) and (F) denotes the clustering coefficient mean (CC) averaged by the 14 electrodes. The y axis in all six figures denotes PRS coherence.

### Subjective affect and 1/f statistics

After nature exposure, participants reported less fatigue and reported more positive emotions than did those exposed to the urban environment. More specifically, after nature exposure, participants reported less increased fatigue (T(30) = 2.45, p = 0.020), more sustained vigor (T(30) = 2.09, p = 0.045), more sustained esteem (T(30) = 2.31, p = 0.028) and less increased TMD (T(30) = 2.88, p = 0.007).

We compared individual global $\beta$ with the POMS affective measures and the Necker change ratio using the Pearson correlation. Significant correlations were found between individual global $\beta$ and their changes in fatigue (cc = −0.410, p = 0.020) and vigor (cc = 0.442, p = 0.011).

### Attention

We did not observe any significant differences between the nature and urban groups in attention changes that were measured by the Necker cube change (p = 0.343), although

such differences have been reported elsewhere (*Tennessen & Cimprich, 1995*). No correlations were found between the Necker cube changes and functional connectivity, either in the 1/f statistic or the small-world network property.

## DISCUSSION

Growing experimental and theoretical evidence supports the notion that human and animal visual systems are adapted to represent natural scenes efficiently (*Simoncelli & Olshausen, 2001*). Because brain vision systems have evolved in the natural world, it has been strongly suggested (*Olshausen & Field, 1996*; *Simoncelli & Olshausen, 2001*) that early visual pathways were adapted to de-correlate the correlational structure of the input signals efficiently and then pool all the encoded information together in the higher level cortex to formulate internal pictures of the outward natural world. Hence, brain sensory systems may function more efficiently in response to a natural scene and consume less energy in the visual processing of natural signals (*Laughlin, 2001*; *Olshausen & Field, 1997*; *Simoncelli & Olshausen, 2001*). As stated in our previous study of non-human primates (*Yu, Romero & Lee, 2005*), the visual function of cortical neurons may be designed to tune to 1/f characteristics for efficient coding, which induces a preference in the visual system for natural signals in the natural environment. Additionally, a recent study (*Torralba & Oliva, 2003*) reported that cardinal (horizontal and vertical) orientations are more prevalent in sample image statistics of man-made artifacts (e.g., urban buildings, streets, highway infrastructures) than in images of natural scenes. Further, there are more low-frequency components and fewer high-frequency components in built artificial views, which result in a larger slope ($\geq 2.5$) in the power spectra of urban images (*Braun et al., 2013*; *Torralba & Oliva, 2003*).

This raises a nontrivial issue. That is, after prolonged exposure to urban artifacts with signal statistics that are distinct from those in nature, could human neural responses differ in terms of statistical properties and therefore become overstressed and uncomfortable (*Penacchio & Wilkins, 2015*)? In fact, recent studies have documented a significant drop in both self-reported rumination and neural activity in the subgenual prefrontal cortex of healthy participants after a 90 min walk in a natural environment, but no such effect was observed following a walk in an urban environment (*Bratman et al., 2015b*).

### Small-world network property and perceived environment coherence

We found enhanced small-world network properties in the brain functional connection during exposure to nature. The functional network in nature was found to be of both a higher functional segregation (larger clustering coefficient, i.e., $C_{mean}$) and a higher functional integration (smaller average shortest path length, i.e., $L_{mean}$) than that of urban exposure. This finding implies that human brains are functionally connected during nature exposure in a way that enables divisions for specialized information processing and coordination between these divisions. This enhanced small-world property has also been observed in the context of music perception (*Wu et al., 2012*) when compared to noise listening.

The small-world characteristics of the functional network during nature experiences were most prominent in the theta band, which is a frequency band that is thought to be associated with new information encoding (*Klimesch, 1999*) and indicates a possible higher efficient network for information encoding in nature. However, additional studies are needed to verify this claim.

Measures of the small-world network characteristics, clustering coefficient ($C_{mean}$) and average shortest path length ($L_{mean}$) were found to be correlated with "coherence," which is a subjective psychological measure that refers to one of the qualities in a restorative environment when one feels that the environment information is in order, is not chaotic and is therefore more comprehensible.

### 1/f statistics, perceived fatigue and vigor

We have also studied the power spectrum properties of EEG signals. The recorded EEG signals also displayed a reversed power law phenomenon known as the $1/f^{\beta}$ statistic. The $1/f^{\beta}$ statistic suggests that long-term correlations exist in the time domain. A large $\beta$ value means that any two time events in the EEG signal have a relatively large correlation time constant. A $1/f^{\beta}$ power spectrum with power tending to diminish with increasing frequency indicates arrhythmic activities with no particularly dominating periodic oscillatory dynamic. EEG signals are more commonly analyzed for rhythmic activity (*Başar et al., 2001*), with arrhythmic statistics discarded as noise. Indeed, the arrhythmic activity may be functional and meaningful during information processing and cognitive tasks (*He, 2011*; *He, 2014*; *He et al., 2010*; *Ray & Maunsell, 2011*).

We observed a larger $\beta$ in the $1/f^{\beta}$ statistics during the nature exposure than during the urban exposure. Additionally, the $\beta$ values in the right hemisphere are generally higher than those in the left hemisphere. Because a large $\beta$ value means that there are more low-frequency components in the EEG signal, this result suggests slower brain activity without strong high-frequency spiking activities. Therefore, a large $\beta$ following nature exposure herein suggests a more relaxed, less task-loaded state with weaker activities as evidenced by brain imaging experiments (*He, 2011*; *Ward, 2002*). This conclusion is consistent with the mental states reported by the participants wherein the global average $\beta$ was positively associated with the affect changes in vigor (corr. beta = 0.442, p = 0.011) and negatively associated with fatigue (corr. beta = −0.410, p = 0.020). However, identifying the best range of $\beta$ values for the brain in the best performance and/or the most pleasant state will require a better designed task that should be investigated in future studies.

### Limitations

This study makes an initial attempt at understanding brain functional connectivity in response to environmental exposure. Although interesting evidence about enhanced functional connectivity in natural environments has been revealed herein, there are a few limitations.

First, this study only revealed the general impact of un-attentional multisensory environmental signals on functional connectivity. The study was designed to understand the holistic in-situ impact of nature and urban environments during a free-exploring

multisensory naturalistic experience. Our intention was not to draw any causal connections between specific environmental stimuli and brain activities. To specify exactly how much different types of signals in the environment impact brain functional connectivity, more strictly controlled experiments with isolated stimuli and both controlled saliency and motion will be needed.

Second, this study examined the detailed impact of exposure to very few nature and urban sites. However, to explore how the results and conclusions could be generalized across various types of nature and urban sites, more extensive control conditions and more sites will need to be examined in future studies. We cannot specify whether the signal statistics, such as the 1/f characteristic, promoted a more efficient information processing functional network. We also do not know whether we accidently selected a less information-loaded nature site, which may have afforded an easier and more fluent cognitive process.

Third, this study used a between-subject design that introduced unnecessary variance. The between-subject design was adopted to minimize possible learning effects; however, the design also introduced more individual variance to the designed comparison. Therefore, a within-subject replicated design would have better controlled for individual variances.

## Implications and future directions

The general observation that brain activities contain long-term correlation and oscillatory components might reflect interactions between bottom-up information processes and top-down cognitive feedback. This conclusion requires further investigation to deepen our understanding of cognitive computation processes within the brain. Constructing large-scale cortical circuit models to investigate the biophysical mechanism of the 1/f response and the oscillation characteristics as revealed in this study will be interesting. The brain response features, which are potentially important to brain functioning (*He, 2014*; *Watts & Strogatz, 1998*), have not been largely considered in current cognitive computational models. Moreover, the preference of the brain sensory system for the 1/f characteristic may also require further investigation of cortical computational modeling to reveal the biophysical mechanism underlying cortically efficient coding. The experimental investigation of the relationship between environmental statistics and cognitive processing may be valuable for numerous reasons. First, such an investigation may deepen our understanding of the functional properties of brain sensory systems. Second, the derivation of new cortical computation models based on environmental statistics for efficient coding may be fostered. Finally, further investigations might also be helpful in the design of new forms of stochastic experimental protocols and stimuli for probing different brain sensory systems.

In summary, our studies suggest that a nature environment, which is characterized by long-term correlation statistics in visual signals, can evoke different brain oscillatory activities and arrhythmic activities that may help place brain functioning in a more restorative experience. Brains generate more memory-like effects of auto-correlated electrical signals when we are visualizing nature than when we are visualizing a busy urban

environment. This result might be an indication of better memory formation in the natural environment than in the urban environment. Moreover, this "redundant" autocorrelation, which is observed for many natural phenomena and neuronal signals, may be important to our resilience and mental well-being; however, this idea will require further investigation.

## SUMMARY

Increasing experimental evidence has reported that animal sensory neurons process natural signals more efficiently than artificial signals (*Lewen, Bialek & Steveninck, 2001*; *Rieke, Bodnar & Bialek, 1995*). Computational studies have revealed that the structure and function of sensory neurons and networks may be designed to efficiently encode signals from natural environments due to long-term adaptation and evolution mechanisms. Our study was designed to examine whether the human brain functions more efficiently in a natural environment than in an urban environment. The answers and underlying mechanisms related to this question may be critical in understanding the operating principle of the human brain.

Previous experiments have revealed that a short walk in a natural wooded area may greatly refresh the brain's cognitive performance and alleviate negative stressors. Our study indicated that the distinct statistical properties of urban artifacts may also shape the human brain response properties differently and may thus further stress human brains. Additional investigations are needed to more precisely examine the performance differences observed between these two types of environments.

This study revealed the presence of a more efficient brain network during a nature experience than during an urban experience. Specifically, stronger global functional connectivity was observed in nature, and a more enhanced small-world property with a larger clustering coefficient ($C_{mean}$) and a smaller average shortest path length ($L_{mean}$) was also observed, which were most prominent in the theta band. The enhanced small-world properties were found to be correlated with a "coherent" experience measured by PRS psychological scale (*Hartig et al., 1997*), referring to an orderly, comprehensible environment experience. The more efficient brain network in nature may help explain the restorative experience reported in this study and may explain the potential cognitive improvements exhibited shortly after a nature experience, as observed elsewhere in the literature (*Berman, Jonides & Kaplan, 2008*; *Bratman et al., 2015a*; *Taylor & Kuo, 2009*).

This study also documented some relevant changes to long-term correlation characteristics of human EEG signals during in-situ environmental exposures to nature and to an urban environment. We found a larger $\beta$ exponent in the $1/f^{\beta}$ frequency spectrum during the nature experience and a larger $\beta$ in the right hemisphere during the exposures to nature and to urban environments. This study may be among the earliest studies of changes in the 1/f power spectrum exponent $\beta$ of human EEG signals as a function of different environmental experiences. A larger sample size and repeated experiments are needed to further investigate the key properties that are intrinsic to the nature environment that impact the resulting EEG signal statistics. Environmental impact

on the brain functional connectivity characteristics observed in this study may be beneficial in constructing better cognitive computational models.

### Funding

This project is funded by the National Natural Science Foundation of China (31271170, 51408429), China 863 program (2015AA020508), the program for the Professor of Special Appointment (Eastern Scholar SHH1140004) at Shanghai Institutions of Higher Learning, and Shanghai Pujiang Program (14PJC099) and Tongji Architectural Design (Group) Co., Ltd. and Key Laboratory of Ecology and Energy-saving Study of Dense Habitat (Tongji University), Ministry of Education (Grant No. 2015KY06). The funders had no role in study design, data collection and analysis, decision to publish, or preparation of the manuscript.

### Grant Disclosures

The following grant information was disclosed by the authors:
National Natural Science Foundation of China: 31271170, 51408429.
China 863 Program: 2015AA020508.
Shanghai Institutions of Higher Learning, and Shanghai Pujiang Program: 14PJC099.
Tongji Architectural Design (Group) Co., Ltd. and Key Laboratory of Ecology and Energy-Saving Study of Dense Habitat (Tongji University), Ministry of Education: 2015KY06.

### Competing Interests

The authors declare that they have no competing interests.

### Author Contributions

- Zheng Chen conceived and designed the experiments, performed the experiments, wrote the paper, reviewed drafts of the paper.
- Yujia He analyzed the data, contributed reagents/materials/analysis tools, prepared figures and/or tables, reviewed drafts of the paper.
- Yuguo Yu conceived and designed the experiments, contributed reagents/materials/ analysis tools, wrote the paper, reviewed drafts of the paper.

### Human Ethics

The following information was supplied relating to ethical approvals (i.e., approving body and any reference numbers):

This study was approved and supervised by the Ethics Committee of Tongji University (no. 2015yxy103).

### Data Deposition

Zenodo: http://dx.doi.org/10.5281/zenodo.55847.

## Supplemental Information

Supplemental information for this article can be found online at http://dx.doi.org/
10.7717/peerj.2210#supplemental-information.

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
