# Peer review of "Enhanced functional connectivity properties of human brains during in-situ nature experience"

_PeerJ, doi:10.7717/peerj.2210_

## Round 0.1 · original submission · Major Revisions

I absolutely agree with reviewer #1 that the idea of getting out of the laboratory into natural and urban environments is a great one. Moreover, I understand that this comes with various challenges. Some of these challenges (matching conditions) may, in a first study, only be rudimentary addressed. However, it should be made very clear, how and in which way these matching conditions have to be improved in future studies and how this could affect your results. Reviewer #3 rises various concerns in this direction.

Please follow all the raised issues of all three reviewers in your re-submission.

·

Basic reporting

Intro and Background:
The introduction emphasizes "social" aspects, but the research is a single person sitting without interaction. There is no basis for asserting the relevance to social psychology.

Data Sharing:
I did not see a link to the data reported anywhere.

Experimental design

Methods should be described...:
The methods are not well described. The attached pdf annotates where I needed additional details and clarity.

Validity of the findings

No comments beyond what is annotated in the pdf.

Additional comments

I really like the idea of getting out of the laboratory into natural and urban environments to evaluate their effects. And I admire the diligence and effort to do eeg on site. But I don't feel the authors will get the attention their effort deserves, because of questions concerning exactly what was done, and how it was done. That can be addressed by clearer writing. The more important issue for the authors is their rather idiosyncratic measure of functional connectivity which was to correlate the amplitudes of the fourier coefficients of the power spectra by EEG band, rather than the measured voltages directly. I think if the reason for choosing the particular measure they did should be better developed, and then paper might have a greater influence. In addition, they should be a lot more methodical in sharing the data processing details so it would be easier for a reader to follow. The graphical presentation of the data could have greater impact if it included some direct visualizations of the change in connectivity. A final comment is that the authors talk about social pressure in the intro and conclusion, but there is no social component of the study. This goes beyond speculation. The authors may be interested in this topic, but this doesn't seem the place to develop it. Rather this is a nice effort to evaluate urban and nature influences on cortical electrical activity. It shows effects, and thus lays some groundwork for the future, which might include the social components.

My markup.
1. Strike-out - things I thought could be removed.
2. Highlights - possible typos
3. Squiggles - indicate the area of text where I have a particular comment. See annotation
4. Text - general comments not closely bound to one particular sentence.

Reviewer 2 ·

Basic reporting

In the present paper the authors investigate impact of exposure to natural or urban environment on brain dynamics measured by EEG. Going in line with previous works they relate functional network measures to different environmental stimuli.

Although this work is interesting, I do have a few misgivings about the paper, set out below.

First of all, the article must be reviewed by a native English speaker to be publishable. Although the meaning can be understood, it is hard to follow. For example, there are dozens errors such as in ‘…during nature exposure compared ‘at’ that during..’ (line 247)

Experimental design

Line 98. An explanation of the experimental design contradicts to what is written in the abstract. How many participants are exposed to the natural and how many to the urban-like environment?
Line 103 - 111. Provide this data in form of a table.
Line 186. Small-world property should be calculated with respect to random reference network with the same network density. Refer to the relevant literature and give correct form of the equation to calculate network small-worldness.
Line 199. The Floyd algorithm is likely unfamiliar to many readers, some additional information on what it tells us would be useful.
Line 206. Give details about the transformation used to calculate spectral characteristics of the signal.
Line 213. Describe all statistical test used in the study and give the rationale for your choice.

Validity of the findings

Line 250. Given the way how small-world coefficient is calculated this section needs to be rewritten accordingly.
Figure 2.
Plot error bars in each graph. Denote significant differences between natural and BE measurements.
Figure 3.
Improve visibility of x and y labels in figure c and b.
Figure 4.
Again denote significant differences between electrodes in the graph.

·

Basic reporting

Chen and colleagues compare EEG signals obtained from human participants either viewing a nature or an urban natural scene. EEG signals are found to follow small-world connectivity properties substantially more so during nature viewing than during urban scene viewing. The authors conclude that natural environments benefit cognitive performance and mental well being.

Experimental design

A particularly interesting feature of this study is that the authors investigate differences in neural signal properties caused by nature or urban scene viewing in-situ. While such an attempt is particularly commendable it comes with a long list of issues that need to be controlled (the authors address a few), however several issues still need further clarification in this manuscript.

Validity of the findings

Major issue 1:
While the authors mention to have matched external variables such as temperature, wind chill and humidity, it is entirely unclear to me how important scene features such as contrast and motion as well as related internal variables such as eye-movements, head-movement or pupil dilation are controlled. If these controls are unavailable they need at least to be discussed in particular how these signals may be different between the two scene types and they may have or may not have caused the observed differences in neural signals. The authors need to provide a figure with all (un)matched properties. I am saying this because it is mentioned in the introduction that other conditions were matched as well while the reader is left wondering which ones these might be. Please be clearer and provide comparative statistics. I am particularly interested whether the urban scene contained more motion such as cars and people passing by which may have caused a differential state of arousal/attention/alertness or more eye-movements etc. which may cause substantially different brain signals under such conditions which are entirely unrelated to the spatial scene statistics, which however the authors claim to be responsible for the observed neural signal differences.
Major issue 2:
The authors conclude that nature (like) environments may benefit cognitive performance and mental well-being however no direct test is presented to support this claim. Since a between subjects design was used one such test would for instance constitute a regression of subject-wise small world property measure against affective measures in either condition. Ideally such inference should be made using a within subject design. The authors may at least acknowledge this shortcoming if such direct claims are made.

Additional comments

Minor issues:
- Please choose one name for the urban scene condition which is used throughout the manuscript. Urban scene I think is most intuitive but I leave it to the authors’ discretion.
- In the methods section the authors claim to compare brain performance, which I believe is misleading. Please clarify what you mean by this or choose another property that describes best what is actually compared.
- Please make sure the figures/panels mentioned in the text match the actual figures (Fig 1b actually means fig 2 I believe).
- The gray shaded colors in figure 5 are too similar, please make the contrast larger.
- Fig 6b should be at least 2 panels. The figure is too crowded.
- What is shown in figure 6c?
- In the discussion the authors speak of ´frequency bands that are correlated with some special brain behavior state´. The authors need to be clearer. What is particularly meant here? Sentences like this are rather frequent throughout the discussion and need clarification. This culminates in the summary in which some sentences need grammar correction.

---

## Round 0.2 · Minor Revisions

The revision improved the manuscript, only a few minor changes are necessary.

·

Basic reporting

All my concerns are adequately addressed by the revision.

Experimental design

All my concerns are adequately addressed by the revision.

Validity of the findings

All my concerns are adequately addressed by the revision.

Additional comments

All my concerns are adequately addressed by the revision.

Reviewer 2 ·

Basic reporting

NA

Experimental design

NA

Validity of the findings

NA

Additional comments

The author made the great effort to implement the changes in the revised version of the manuscript.

Minor comment:
Line 37: Please refer to the proper figure. I believe it should be Figure 6.

·

Basic reporting

much clearer now

Experimental design

Authors have sufficiently explained the design and its shortcomings

Validity of the findings

Findings are clearly communicated and appear much more conclusive now

Additional comments

In my view the correleations between neural scores and perceived experience are crucial in this ms. For that reason the correlation figure should be included and the interpretation and discussion of these results should be extended.

I believe the manuscript would generate more impact if it featured the term 'in-situ' in the title.

Fig1 x and y axis labels contain cut-off letters and legend in 1a overlaps with yticklabels in 1b

In F7b x-axis: 'Paterial' should be 'Parietal'

---

## Round 0.3 · accepted · Accept

The current version is now accepted for publication.